# Lack of long-term acclimation in Antarctic encrusting species suggests vulnerability to warming

Melody S. Clark [1], Leyre Villota Nieva[1,2], Joseph I. Hoffman [3], Andrew J. Davies [4], Urmi H. Trivedi[5], Frances Turner[5], Gail V. Ashton [6] & Lloyd S. Peck [1]

Marine encrusting communities play vital roles in benthic ecosystems and have major economic implications with regards to biofouling. However, their ability to persist under projected warming scenarios remains poorly understood and is difficult to study under realistic conditions. Here, using heated settlement panel technologies, we show that after 18 months Antarctic encrusting communities do not acclimate to either +1 °C or +2 °C above ambient temperatures. There is significant up-regulation of the cellular stress response in warmed animals, their upper lethal temperatures decline with increasing ambient temperature and population genetic analyses show little evidence of differential survival of genotypes with treatment. By contrast, biofilm bacterial communities show no significant differences in community structure with temperature. Thus, metazoan and bacterial responses differ dramatically, suggesting that ecosystem responses to future climate change are likely to be far more complex than previously anticipated.

[1] British Antarctic Survey, Natural Environment Research Council, High Cross, Madingley Road, Cambridge CB3 0ET, UK. [2] School of Ocean Sciences, Bangor University, Menai Bridge, Anglesey LL59 5AB, UK. [3] Department of Animal Behavior, University of Bielefeld, Postfach 100131, 33615 Bielefeld, Germany. [4] University of Rhode Island, Department of Biological Sciences, Woodward Hall, 9 East Alumni Avenue, Kingston, RI 02881, USA. [5] Edinburgh Genomics (Genome Science), Ashworth Laboratories, Charlotte Auerbach Road, The King's Buildings, The University of Edinburgh, EH9 3FL Edinburgh, UK. [6] Smithsonian Environmental Research Center, 647 Contees Wharf Road, Edgewater, MD 21037-0028, USA. Correspondence and requests for materials should be addressed to M.S.C. (email: mscl@bas.ac.uk)

Understanding why different species thrive or fail under changing environmental conditions is crucial for predicting future biodiversity. In terms of underlying mechanisms, acclimation and physiological flexibility have been proposed as the two main routes dictating the success or failure of species under future climate change scenarios, particularly for long-lived species[1]. This is based on the premise that genetic adaptation in species with long generation times and extended lifespans will not be rapid enough to enable animals to cope with the current rate of climate change[1–3]. Thus, knowledge of the extent of phenotypic plasticity within populations and the capacity of genotypes to produce different phenotypes in response to environmental change is crucial in predicting how global biodiversity will be affected by future warming[1–3]. However, obtaining knowledge of such mechanisms is particularly problematical in long-lived species, such as polar marine invertebrates. Their low metabolic rates and low-energy lifestyles mean that ecological observations of persistence in a warmer world do not necessarily correlate with fitness and population sustainability[2,3]. Long-term experimental manipulations of these species is difficult due to the extended timescales (of years) needed for each chronic exposure experiment, as well as the difficulty in accurately reproducing the biotic and abiotic variables which occur in the natural environment over long periods.

Encrusting communities are ubiquitous and major contributors to benthic biodiversity and community dynamics globally[4–6]. Their ability to rapidly colonise hard substrata also has serious economic implications in terms of biofouling[7]. Hence, there is a major interest in how this community will perform under future climate change scenarios. However, such evaluations are now possible on encrusting (filter feeding) species due to the recent development of in situ heated settlement panels[8]. This technology enables us to heat the thin surface layer of water (up to 5 mm) above the panels to +1 and +2 °C above ambient temperature, matching the 50 and 100 years predictions for warming in the Southern Ocean respectively[9]. This system simulates oceanic warming predictions in the natural environment while maintaining natural cycles of temperature variation, light regime and food supply. The advantage of their use with polar species is that because these cold-adapted species grow very slowly (on average 5× slower than temperate species[3]), even after two or more years, they do not outgrow the heated water layer. Hence, deployment of these panels in the polar regions provides an unparalleled opportunity to conduct ecologically relevant long-term studies and investigate the molecular mechanisms underpinning responses to chronic warming in threatened sub-zero marine ecosystems.

The Antarctic Peninsula is a region of the globe that has experienced some of the most rapid rates of regional climate warming over the past 50 years[10]. Although there are indications that the atmospheric warming may have ceased, there is no evidence to suggest that this trend is reflected in oceanographic data, which reveal ongoing reductions in annual sea ice and glacier retreat[11,12]. This has significant implications for the marine biota, especially the filter feeders[13]. The reduction in sea ice alters water stratification and mixing, which directly affects the strength of the summer phytoplankton bloom[11]. There is also evidence that warming is increasing the proportion of nanophytoplankton in the bloom and therefore changing food availability[11,14]. The Antarctic marine environment is highly seasonal and the long Antarctic winter results in a significant and prolonged reduction in phytoplankton[15]. Filter feeders need to ensure sufficient accumulation of food stores during the relatively short summer season to survive this famine through to the next bloom period[15,16]. Given the changing bloom conditions, filter feeders are more likely to be affected in the near future compared to non-

filter feeding species (detritivores, carnivores etc) and are thus useful indicator species for the effects of climate change.

The initial 2017 heated settlement panel study[8] demonstrated the massive impacts of warming on the encrusting marine assemblages, with a near doubling of growth rates of the encrusting species. This was a very surprising result, especially for such slow growing Antarctic species. Although the experiment was terminated at the end of the Austral summer, there were indications that animal growth rates were starting to slow alongside the decline in the phytoplankton bloom. This left open the questions of whether the encrusting species had acclimated to the new conditions in the long term and, in particular, if they had built up sufficient energy reserves to survive the dearth of their food supply over the long Antarctic winter and maintain their elevated metabolic rates in subsequent years.

In this study, heated settlement panels were deployed for 18 months in Ryder Bay near Rothera Research Station, Antarctica. The prolonged timescale of the experiment enabled studies to encompass the critical winter period, described above. Multiple, complementary aspects of the responses of the settlement panel communities to warming were evaluated, with responses to warming assessed in both encrusting filter feeders and microbial biofilm communities. In this study, we specifically used the responses of spirorbids (calcified marine worms) as model species and proxies for the other encrusting community species on the panels. This was because they were not only present in sufficient numbers and of a sufficient size for the analyses undertaken, but they also showed the same response to warming as the other spatially dominant species on the panels in the original 2017 deployment[8]. Panels comprised controls (non-heated, experiencing ambient temperatures roughly between −2 and +1 °C[17]) and two sets warmed to +1 and +2 °C above ambient sea water temperatures (referred to subsequently as +1 and +2). The health of the spirorbid species under warming was investigated using an RNA-Seq approach and expression profiling. Furthermore, acute Upper Thermal Limit (UTL) experiments were used to determine if physiological acclimation had occurred at the whole animal level. The RNA-Seq data were also interrogated for single-nucleotide polymorphisms (SNPs) to investigate differential survival of genotypes between treatments using a population genetics approach. Microbial biofilms can significantly influence community composition[18] and with their very short life cycles would be expected to show more variation in community composition with temperature. Therefore, to provide a contrast to the slow growing, slowly developing metazoan encrusting species, prokaryotic bacterial communities on the panels were also evaluated using amplicon sequencing of bacterial 16S rRNA.

Here, we report the results of our 18-month study, which represents the longest acclimation trial on any Antarctic species to date. To our knowledge, it includes some of the longest in situ experimental manipulation of temperature anywhere in the oceans to date (9–18 months). Previous acclimation trials have revealed the very long timescales required for this process in Antarctic marine species[19], but this study reveals the extraordinary sensitivity of the filter-feeding encrusting species to even +1 °C of warming. The small size and solid calcified exoskeletons of the spirorbids precludes the determination of animal health using visual observation, but molecular analyses using RNA-Seq show that they are unable to sustain cellular homoeostasis and are likely in a process of extended decline. Population genetic analyses also reveal little evidence for differential selection of genotypes with temperature. UTL analyses, furthermore, show no increase in UTL in the spirorbids on the warmer plates and therefore no acclimation at the whole animal level. There is no significant difference in the composition of bacterial biofilm communities with temperature and hence no impact of warming.

This suggests that marine metazoan invertebrate and prokaryotic responses to warming are very different, which in turn implies that ecosystem responses to ongoing climate change are likely to be far more complex than previously anticipated.

## Results

**Expression profiling of *Protolaeospira stalagmia*.** The cellular responses to warming of the endemic biota on the heated panels were analysed via RNA-Seq expression profiling of the spirorbid (*P. stalagmia*). Assembly of the raw reads from all samples produced over five million contigs greater than 100 bp that contained over 4.28 Gb of data (Supplementary Table 1). A series of filtering steps (described in Transcriptome methods) provided a reference transcriptome containing 61,421 high-quality transcripts with the mean length of 486 bp and a total size of 29.92 Mb. Of these transcripts, 23.86% contained putative protein sequences, out of which 63.45% showed matches to known proteins from the Swiss-Prot database and 33.92% possessed functional information based on GO terms. While comprehensive searches were conducted using different sequence databases (e.g., Pfam, SignalP, TmHMM, eggnog, KEGG and GO), the most comprehensive annotations were provided by the Swiss-Prot database (BlastX and BlastP).

Differential expression analysis was then performed on the +1 and +2 samples, using the control samples as a baseline in both cases. In the +1 sample versus control analysis, a total of 14,631 transcripts were significantly differentially expressed. The majority of these transcripts (13,034) were up-regulated in the +1 samples compared with the controls. Fewer transcripts were differentially expressed between the +2 samples and the controls (1020 in total) of which 1013 were up-regulated in the +2 samples and only 7 down-regulated. Principal component analysis (PCA) on the normalised and filtered expression data revealed there was significant separation between each set of samples (control, +1, and +2), with the first principle component explaining approximately 57% of the variance of the data ($P = 0.004$) (Fig. 1). The PCA plot shows that while there is good separation between the +1 samples and both the other groups, one of the +2 samples clusters with the control samples, which may explain why fewer transcripts were differentially expressed in the +2

treatment compared to controls. GO enrichment analyses were performed to identify if specific metabolic pathways or functional groups were preferentially expressed in a particular treatment. These analyses showed significant enrichment below the threshold of the false discovery rate (FDR) of 0.05 for only the control versus +2 comparison, which meant that these processes were up-regulated and enriched in the +2 samples. There was a significant reduction in representation of GO categories involved in transcription (Molecular Function: rRNA binding and DNA directed 5′–3′ RNA polymerase activity) in the control samples compared with the +2 samples (Supplementary Table 2).

Thousands of transcripts were up-regulated in the +1 treatment (14,631, or approximately 25% of the transcripts identified in this study), which indicated that even with +1 °C of warming, the animals had to significantly alter their cellular processes and cellular physiologies to accommodate to the new temperature. Given the lack of any GO enrichment for particular categories of gene functions and the large number of transcripts involved, more detailed analyses concentrated on the expression profiles of the +2 samples examining the annotation of the up-regulated transcripts. Of the 1013 transcripts up-regulated in the +2 animals (compared with controls), 312 were annotated via BlastX and BlastP searches (Supplementary Data 1). Most striking about this annotated gene set was the large-scale up-regulation of transcription and translation processes, comprising almost one-third of transcripts. The majority of these sequences were ribosomal genes, but elongation and translation factors were also present. These results corroborate the GO enrichment analyses described above (Supplementary Table 2). Similarly there was up-regulation of transcripts putatively involved in cell division (e.g., G2/mitotic-specific cyclins, proliferation proteins), histones, which play major roles in chromatin organisation and regulation, transcripts for protein degradation (e.g., E3 ubiquitin ligases), and cellular respiration (e.g., cytochrome *c* oxidases). A number of classical stress response transcripts were also up-regulated including numerous heat-shock proteins and other chaperone transcripts coding for T-complex proteins and peptidyl-prolyl *cis-trans* isomerases (Supplementary Data 1). To identify critical biochemical pathways, the STRING programme was used to visualise protein–protein interactions and analyse enrichment. This uncovered statistically significant enrichment of certain

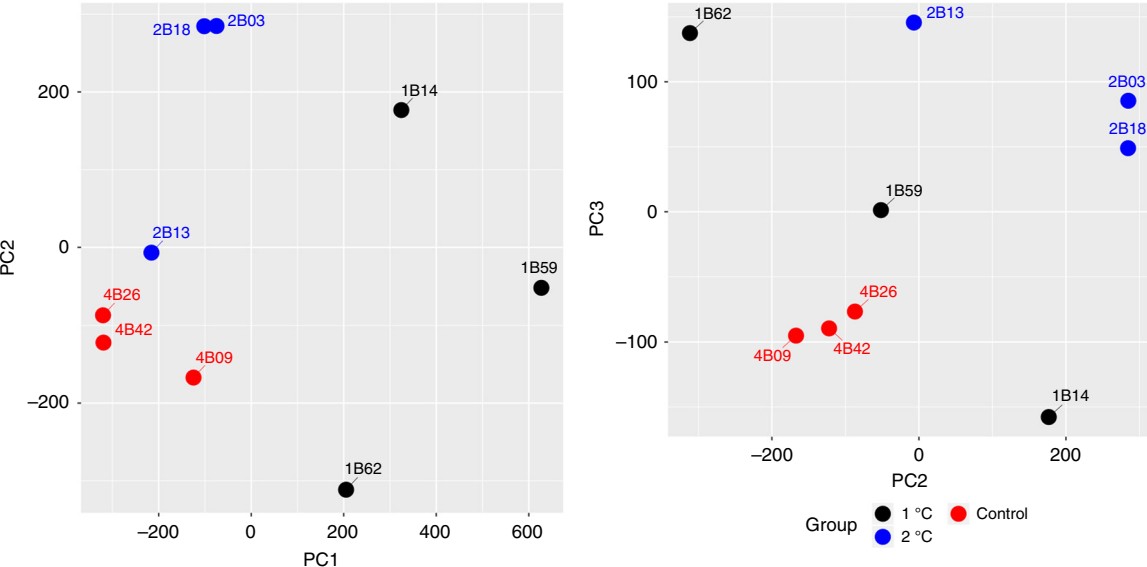

**Fig. 1** PCA plot on normalised and filtered expression data of the different treatments. **a** First and second components of the PCA. **b** Second and third components of the PCA. Source data are provided as a Source Data file

**Table 1 PANTHER v13.1 GO-slim overrepresentation tests for _P. stalagmia_**

| Process | GO identifier | FDR |
|---|---|---|
| Biological processes for +2 °C up-regulated transcripts | | |
| Translation | 0006412 | $2.02e^{-13}$ |
| rRNA metabolic process | 0016072 | $4.05e^{-06}$ |
| Protein folding | 0006457 | $8.62e^{-03}$ |
| Generation of precursor metabolites and energy | 0006091 | $4.05e^{-03}$ |
| Cellular component biogenesis | 0044085 | $7.04e^{-10}$ |
| Cell cycle | 0007049 | $2.90e^{-04}$ |
| Organelle organisation | 0006996 | $8.26e^{-07}$ |
| Biosynthetic process | 0009058 | $8.95e^{-06}$ |
| Molecular function for +2 °C up-regulated transcripts | | |
| Structural component of ribosome | 0003735 | $2.57e^{-42}$ |
| Translation elongation factor activity | 0003746 | $7.32e^{-03}$ |
| Translation initiation factor activity | 0003743 | $4.55e^{-02}$ |
| Translation regulator activity | 0045182 | $5.76e^{-04}$ |
| Hydrogen ion transmembrane transporter activity | 0015078 | $3.46e^{-03}$ |
| Structural component of cytoskeleton | 0005200 | $2.23e^{-08}$ |
| Nucleotide binding | 0000166 | $6.70e^{-05}$ |
| mRNA binding | 0003729 | $2.44e^{-04}$ |

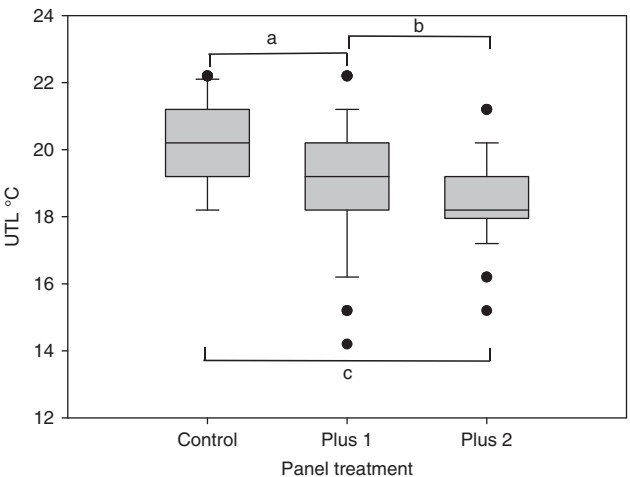

**Fig. 2** Box plot showing acute UTLs for the spirobid polychaete _R. perrieri_. Central line of each box denotes the median, the top and bottom edges of the box show the 25% and 75% percentile, with the 10% and 90% percentiles shown by the upper and lower whiskers; outliers are shown as circles. Letters denote statistically significant differences between treatments (a: $P = 0.0129$; b: $P = 0.000$; c: $P = 0.0200$) (Mann–Whitney pairwise comparisons). Source data are provided as a Source Data file

**Table 2 SNPs, genes and GO terms found to correlate with temperature in _P. stalagmia_**

| Analysis results | Number |
|---|---|
| Number of SNPs | 13,843 |
| Number of SNPs with $Z$-score >0.4999 | 839 |
| Number of SNPs with FDR ≤ 0.05 | 0 |
| Number of genes with ≥5 SNPs | 521 |
| Number of genes with FDR ≤ 0.05 | 91 |
| Number of GO terms with FDR ≤ 0.05 | 0 |

functional groups of proteins ($P < 1.0e^{-16}$), with a major cluster of transcription and translation proteins and satellite clusters of proteins involved in post-translational modification, the cell cycle, cytoskeleton and energy production (Supplementary Fig. 1). PANTHER enrichment results, of the same dataset, were dominated by GO terms associated with RNA metabolism and cell division (Table 1). Overall, these data indicated the induction of the classical stress response to the warmer conditions and a lack of acclimation. The latter was further tested via acute UTL analysis.

**UTL analysis of _Romanchella perrieri_.** Whole animal acclimation is traditionally tested using acute UTL analyses. Due to the limited representation of _P. stalagmia_ on the panels, a closely related spirorbid species (_R. perrieri_) was used to test UTLs associated with treatment, with animals taken from the same sets of panels, as were used as a source for the _P. stalagmia_ and the transcriptome experiments The UTL data supported the _P. stalagmia_ transcriptome analyses, with a lack of whole animal acclimation in _R. perrieri_ and a reduction in thermal resilience associated with increased panel temperature (Fig. 2). A Kruskal–Wallis test confirmed that there was a statistically significant effect of panel treatment on the UTL of _R. perrieri_ ($H = 26.65$, $DF = 2$, $P = 0.000$) and that mean UTL of the animals declined with panel temperature (control = 20.20 °C ± 0.167 SE mean, +1 panels = 19.26 °C ± 0.269 SE mean and +2 panels = 18.62 °C ± 0.167 SE mean). Mann–Whitney pairwise comparisons revealed statistically significant differences between all the treatments (control vs. +1 $P = 0.0129$; control vs. +2 $P = 0.000$; +1 vs. +2 $P = 0.0179$) (Fig. 2). Thus the UTL declined in the warmed individuals, contributing to evidence of a stress response, as indicated in the molecular results above.

**Population genetic analysis.** In order to test for allele frequency differences among the treatments, a total of 13,843 SNPs were called from the RNA-Seq data according to strict criteria (see the Methods for details). Analysing each SNP individually resulted in no significant allele frequency differences among the groups after FDR correction (Table 2). However, when the mean $Z$-scores of SNPs across genes were assessed, 91 out of 521 genes (17.5%) showed a significant association with temperature after FDR

correction (Table 2). It should be noted that these 91 genes comprised only 0.14% of the total number of transcripts obtained in this study. GO annotations could only be recovered for 14 of these genes (Supplementary Table 3). Blast matches were obtained against ribosomal sequences, cytoskeletal proteins and serine/threonine kinases (Supplementary Table 3). The former are involved in translation, a result that corroborated the transcriptome and GO enrichment results. Cytoskeletal proteins such as tubulin and actin are structural proteins that are often involved in the cellular stress response, while serine/threonine kinases are proteins that play critical roles in signal transduction affecting cellular processes such as cell division, proliferation and apoptosis.

**Biofilm oligotype analyses.** Finally, to compare the responses to warming in eukaryotes with prokaryotes, the panel biofilm communities from the different treatments (control, +1 and +2) were characterised using 16s amplicon sequencing. The results were subjected to Oligotyping analysis, which uses Shannon entropy[20] to identify positional variation in order to facilitate the identification of nucleotide positions of interest[21]. This enables the detection and classification of distinct subpopulations within a genus or even within a single species as shown for _Gardnerella vaginalis_ in humans[22]. This technique does not rely on reference databases and clustering approaches to identify operational taxonomic units, which can often be problematical, as the complete reference genomes of known environmental organisms are

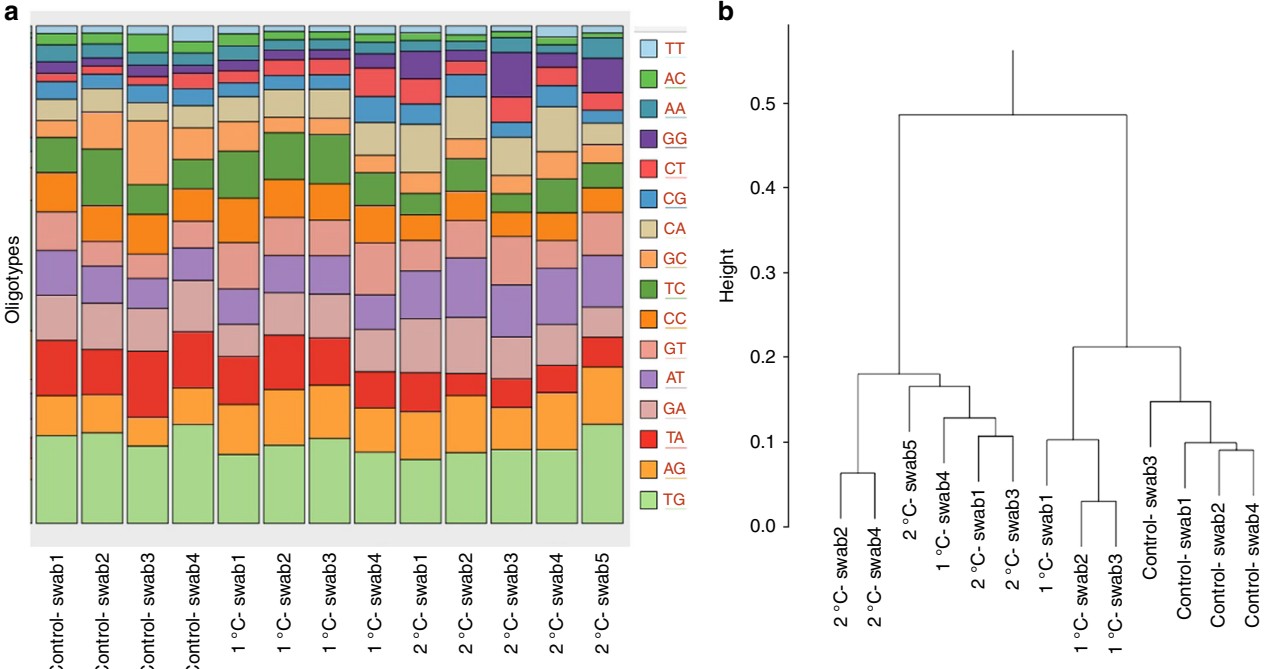

**Fig. 3** Oligotype analyses identifying the most abundant biofilm. **a** Proportions of the 16 abundant oligotypes in the biofilm samples for the different treatments (control, +1 and +2). Source data are provided as a Source Data file. **b** Bray–Curtis distance measures for oligotypes from biofilm communities from panels (controls, +1 and +2)

often lacking in the databases[23,24]. This is even more predominant in Antarctic marine biofilms that are under-studied[25]. Oligotyping results yielded 16 unique highly abundant oligotypes (>1% relative abundance, comprising 90% of all reads from a total of 1,478,129 sequences) across all 14 samples. The abundance of each highly abundant oligotype varied across the treatments, although not significantly (PERMANOVA, $P = 0.16$) (Fig. 3a). The two most abundant oligotypes were TG (abundance of 12.8–19.9%) and AG (abundance of 5.8–11.5%) (Fig. 3a), but these abundances were not significantly different between treatments (PERMANOVA, $P = 0.2$). Cluster analysis using Bray–Curtis similarity ratios obtained from normalised oligotype data revealed no biofilm community differences between the +1 and control treatments with both clustering together (Fig. 3b). The exception was one +1 sample which clustered with the +2 samples. The +2 community composition was more variable and different to the +1 and control treatment, with the exception of the single swab from the +1 community.

In the oligotyping analysis, the minimum relative abundance threshold removed 350 rare oligotypes. However, given the lack of any association of the abundant oligotypes with temperature, further analyses were performed on these rare oligotypes to identify if they were more frequent with, or associated with, a particular treatment. These rare oligotypes were similar in the presence and abundance across treatments. There were two exceptions that were slightly different across treatments in terms of percentage abundance. Rare oligotype 1 (GAGGTTAGTCTG ATCTGGGACCCAACGT) was more abundant in the +2 samples (ranging from 17.7% to 25.6% of rare oligotypes) than in the +1 samples (5.9% to 7.7%) and absent in the control treatment. The abundance of rare oligotype 2 (AGGTTGTTAACAGCT GAATCAACTG) was similar across treatments except in two of the +2 samples (17.2% and 25.6% of rare oligotypes) where it was more abundant than in any of the other samples and treatments (ranging from 0.2% in one control sample to 13.5% in other +2 samples).

**Assignment of biofilm taxonomy**. This was carried out using BLAST sequence similarity searching. All of the sequence matches comprised uncultured marine bacteria and were often associated with biofilm studies. The majority of oligotypes were assigned to the phylum level, mainly representing Proteobacteria, with three (GA, AG, TG) matching members of the phylum Bacteriodetes and a single match to Verrucomicrobia. Only four oligotypes were assigned to the genus levels (TT, GC, TC, AG) of *Granulosicoccus*, *Leucothrix*, *Shewanella* and *Saprospira*, respectively. Antarctic studies are poorly represented in the databases and this was reflected in the fact that only two oligotypes (TT, CG) matched sequences from previous Antarctic studies, while an additional oligotype (TA) showed sequence similarity to data from an Arctic study (Supplementary Table 4). Rare oligotype 1 revealed closest sequence similarity to *Leucothrix* spp (accession number: HG934341), a gammaproteobacteria epiphyte isolated from a red algae in Chilean waters, while rare oligotype 2 most closely matched an uncultured marine bacteria of the gammaproteobacteria class (accession number: LC171298).

## Discussion

It has been extensively documented that Antarctic marine species (metazoans) are highly stenothermal with extended periods of up to 8–9 months required for acclimation[3]. These previous evaluations were conducted in the laboratory and the experimental conditions could not capture natural variability, including complex abiotic and biotic interactions[26]. However, the sensitivities identified in the laboratory are re-enforced by this in situ experiment, with both expression profiling and acute UTL analyses showing a lack of acclimation in the spirorbids.

The composition and growth performance of Antarctic encrusting communities on heated settlement panels have previously been described[8]. In the initial study, results demonstrated massive impacts of warming on the encrusting marine assemblages[8]. The most surprising result was the near doubling of

growth rates of the encrusting species, in particular the two main space colonisers the bryozoan *Fenestrulina rugula* and the spirorbid *Romanchella perrieri*[8,27,28]. The results were highly species-specific. On the +1 panels, colonies of *F. rugula* were over double the size of those on control panels, while *R. perrieri* increased in size by 70%. On the +2 panels, the colonies of *F.rugula* were larger than those on control plates, but smaller than those on the +1 panels. In contrast *R. perrieri* showed similar growth rates at both warming temperatures, while a second spirorbid *P. stalagmia* increased in size by a further 20–30% compared with +1 animals. In terms of percentage cover and abundance, the presence of *R. perrieri* was reduced with warming (7.5%, 2.0%, 1.26%: control, +1, +2 respectively), whereas that of *P. stalagmia* was less so (4.5%, 3.25%, 3.5%). These elevated growth rates were maintained throughout the Austral summer (December–February). In March, when the ambient temperature started to fall and the phytoplankton bloom decayed (leading to a reduction in food for these filter feeders) growth rates changed, but again these varied according to species. The growth rates of *F. rugula* on the heated panels declined faster than those on control panels; those of *R. perrieri* showed little difference between treatments while growth rates of *P. stalagmia* continued to increase[8]. This experiment was terminated at the end of the Austral summer. This left open the question of whether these species had properly acclimated to chronic warming and built up sufficient energy reserves during the summer (which was also their period of most rapid growth) to survive the dearth of their food supply over the long Antarctic winter and maintain their elevated metabolic rates in subsequent years.

The 2017 study[8] purely focussed on survival, community composition and growth as area covered, which are not informative about the acclimation status of the animals. This is particularly true of the encrusting communities on the panels, which largely comprised bryozoans and spirorbids. These are small animals with highly calcareous exoskeletons, which are relatively robust and will persist long after the death of the animal[29]. Thus, expression profiling of a sub-set of individuals is a useful method for assaying animal health and physiological state at the cellular level[30]. In this experiment, the responses of the spirorbid worms were used as models for the other spatially dominant filter feeding species on the panels. Expression profiling of the +1 animals revealed a highly active response to the warmer conditions with thousands of genes up-regulated compared to control animals. Relatively little differential expression would be expected between these two treatments, if acclimation had occurred, with the physiology of the +1 animals being re-set to that of the controls. Thus, the +1 animals were still unable to acclimate their physiologies to the warmer conditions, even after 18 months.

At the higher temperature, analysis of the annotations associated with the up-regulated transcripts in the +2 animals revealed indications of cell stress and continued resistance to the warmer conditions. Almost one-third of the transcripts comprised ribosomal genes. Similarly, there was up-regulation of transcripts putatively involved in translation (e.g., translation elongation factors), protein degradation (e.g., ubiquitin and proteosome transcripts), cellular respiration (e.g., ATP synthase) and cell division (e.g., G2/mitotic-specific cyclin), indicating a substantial requirement for the enhanced generation of proteins, protein turnover and cell renewal in the warmer conditions (Supplementary Tables 2 and 3). Critically a number of cytoskeletal proteins (e.g., actin, tubulin), proteins were also up-regulated. These proteins are increasingly being shown to play roles as stress sensors in marine invertebrates[31,32] and there was also evidence of activation of the classical cellular stress response[33]. This was represented by the presence of chaperone transcripts for the 70 and 90 kDa heat-shock proteins (HSP70s

and HSP90s), T-complex proteins and antioxidants such as aldehyde dehydrogenase. This transcriptional profile indicates that the animals were exhibiting cellular stress, having serious difficulties with acclimating to the new conditions and potentially at, or close to, a tipping point in their abilities to survive[34]. These expression profiles are incompatible with those expected of natural senescence, which would be typified by a general down-regulation and would not invoke the cellular stress response. Antarctic spirorbids can live 4–5 years (David K. A. Barnes, personal communication 2019), which is much longer than spirorbids in other regions of the planet. Spirorbids on control panels retrieved after 2.5 years were still alive and reactive (L. S. Peck, personal communication 2019). Thus, even with potentially accelerated ageing in the warm conditions, the timescale of these experiments was well within the natural life span of these animals. While the temperature tolerances of the spirorbids in the different treatments could be affected by body size, all of the animals tested were mature adults. Previous work indicates that there is likely to be a very small effect of body size in mature adults[35] and that this would not influence the results. The expression profiles indicate that while the spirorbids can grow rapidly under warming, they cannot sustain this rate indefinitely. This rapid growth may be a particular issue for filter feeders, where they have access to abundant food during the short Austral summer, but then have to maintain these enhanced metabolisms during the long winter, when phytoplankton are largely absent.

The molecular data were supported by the acute UTL trials, which were conducted on a closely related spirorbid species (Fig. 2). Unfortunately because of the limited numbers of *P. stalagmia* individuals available, all the acclimation experiments could not be carried out on the same species. However, *R. perrieri* is closely related to *P. stalagmia* and fills the same ecological niche within this community. If the spirorbids had acclimated at the whole animal level, this would have been shown as an increase in their UTL[19,35,36]. This did not occur and all animals on the warmer panels were in a permanent state of resistance and/or decline depending on the individual and temperature. The fact that this was still in process after 18 months exposure was probably a reflection of the low-energy lifestyle of these species and their very slowed metabolisms[3]. This highlights a potentially critical factor in understanding Antarctic marine species responses to climate change, that of extended decline. Such a process can only be evaluated using extremely long evaluations combined with molecular techniques, as persistence with decline may not be accompanied by outward morphological or even physiological signs in such long-lived, relatively inactive cold-adapted species[37].

Thus, even a +1–2 °C increase in the environment of these Antarctic encrusting species significantly impacts on cellular homoeostasis, a situation that is likely not sustainable long term, especially with regard to the ability of these animals to maintain enhanced growth rates and sufficient seasonal food stores to ensure longevity. This is in contrast to the previously longest exposure of 17 months to +2 °C[38]. In that study the sea urchin *Sterechinus neumayeri* was exposed to the combined stresses of temperature and low pH in laboratory conditions and showed no significant physiological effects. In fact, reproductive capacity improved with time[38]. However, *S. neumayeri* is a scavenger/detritivore and was fed constantly during the course of the experiment. That experiment further kept animals at constant temperatures, as do nearly all laboratory studies, and lacked the effects of winter seasonality. Our results suggest that the various feeding guilds may respond differently to climate change depending on access to food and/or that winter conditions and seasonality may have a role to play in species persistence.

Given the lack of acclimation in these animals, it was of interest to identify if there had been any differential survival of genotypes

in relation to temperature. Bayenv2 was chosen as the analysis tool because it can identify associations between allele frequencies and environmental variables while accounting for sampling error in pooled data[39]. Small but significant allele frequency differences were observed in a total of 91 genes (0.14% of the total number of transcripts obtained in this study) (Supplementary Table 3). Blast searching and GO analysis of individual genes produced few annotations, but some transcripts had putative functions associated with translation, the cytoskeleton and serine/threonine kinases (Supplementary Table 3), which linked directly with the major transcriptome results above. Overall, there was little evidence for differential survival of genotypes within the spirorbid population. These molecular and UTL results suggest a lack of physiological and genetic flexibility in the encrusting species to warming. The panels were clean when they were put into the water and then warmed up, during which colonisation took place. Hence, phenotypic plasticity could impact several stages, such as post-settlement development, early juvenile development and adult maturity. It is not possible under this experimental design to determine where phenotypic plasticity was constrained, but the overall result was a lack of physiological flexibility in the mature population.

To provide a contrast to the long-lived metazoan species, the bacterial biofilm communities (with very short generation times) were also evaluated on the heated panels. Biofilms are critical components of most marine systems and provide biochemical cues that can significantly impact overall community composition[18]. Previous molecular analyses of biofilms cultured under different environmental conditions have shown dramatic alterations in community composition[40,41]. However, these experiments used short exposures of between 7 and 18 days and relatively high temperature increases (+3 and +5 and 6, 12 and 18 °C)[40,41] that have little relevance for climate change responses. Thus, they may not represent mature biofilm communities and the thermal responses tested were to conditions well in excess of IPCC medium-term scenarios[9]. The data shown here represent more realistic predictions of the future responses of biofilms to climate change. Even though +1 and +2 °C above the ambient freezing temperatures of the Southern Ocean represented proportionally large increases in temperature, when compared with the same temperature increases applied to temperate or tropical systems, there were no significant changes in bacterial biofilm community composition with temperature as revealed by amplicon sequencing. This is in contrast to differences observed in the metazoan community assemblage at 9 months[8]. This indicates a potentially large flexibility within bacterial physiologies to cope with changed conditions, but may also be due, at least in part, to their very short-generation times and therefore potential for adaptation and evolution in much shorter time periods.

Similar to the transcriptome analyses, there was limited annotation of the bacterial species. In this study, many of the oligotypes produced highest sequence similarities against uncultured marine bacteria (Fig. 3) of the Gammaproteobacteria class, which was similar to the results obtained from benthic Antarctic marine biofilms around McMurdo station[25]. However, improved sampling and annotation since the 2006 study[25] did result in reassuringly high sequence similarities with two Antarctic and one Arctic metabarcoding study[42–44]. In the Antarctic, there is a lack of knowledge about most bacterial biofilms and their response to environment conditions. Of the few studies performed, results have indicated that Antarctic marine bacteria can thrive in wider temperature ranges when compared with the invertebrates. For example, a *Pseudoalteromonas* species from the South Shetlands demonstrated growth over a temperature range of −2 to 18 °C, while a *Cellulophaga* species from the same area

survived up to 41 °C[45], which underpins the findings of bacterial thermal flexibility identified here. This finding also consolidates the hypothesis of greater resilience and adaptability of prokaryotic communities under future climate change compared with the high thermal sensitivity of the *Metazoa*. Thus, ecosystem responses to future climate change are likely to be far more complex than previously anticipated.

## Methods

**Panel sample collection.** Samples were taken for molecular analyses from heated (+1 and +2 °C above ambient sea water temperatures) and non-heated (control) panels after 18 months immersion at 15 m depth near Rothera Research Station, Adelaide Island, Antarctic Peninsula (67°4′07″S, 68°07′30″W). These treatments are referred to as control +1 and +2 in the following methods. At the start of the 18-month deployments, all panels were brand new, placed on site and then gradually warmed up to the relevant temperature for colonisation in situ. Three sites around Rothera were used for the original study, with panels deployed in South Cove, Hangar Cove and North Cove on concrete substrata. At each site, four heated panels for each temperature (total of 12 panels per site) were laid in a random design in batches of four with position generated by a random number generator and this design was random with regard to both the position and the concrete block. Because of iceberg impact damage, one set of panels was retrieved from South Cove after 9 months and held in the Rothera flow-through aquarium. The water intake for the aquarium is not filtered; the inlet is at 7M and only boulder covered to protect from ice. Therefore, the animals in the aquarium experienced the same sea water, food availability and environmental conditions as the original site for a further 9 months (South Cove panels). It should furthermore be noted that most of this final 9-month period took place during winter when phytoplankton levels are extremely low. Another set of panels (North Cove panels) remained in the sea for the whole 18-month period. It was not possible to use the Hangar Cove panels in this study, as predation by urchins had dramatically altered the community composition of the panels at this site. Experimental organisms were non-regulated so ethical approval was not required. After a preliminary environmental assessment (PEA #14-11) collections were made within the Antarctic Act, permit number S7-10/2015, as granted under section 7 of the Antarctic Act 1994.

**Transcriptome methods.** *Protolaeospira stalagmia* were dissected from their calcified skeletons, snap frozen in liquid nitrogen and stored at −80 °C prior to RNA extraction. Total RNA was extracted from dissected *Protolaeospira stalagmia* ($n$ = 6 per treatment: 3 each of control, +1, +2; total of 54 individuals) using ReliaPrep TM RNA Miniprep Systems (Promega) according to the manufacturer's instructions. RNA samples were assessed for concentration and quality using a NanoDrop ND-100 Spectrometer (NanoDrop Technologies) and an Agilent 2200 Tapestation (Agilent Technologies). RNA samples ($n$ = 6 per panel, per treatment) were pooled to obtain a total of three replicates per treatment (control, +1, +2) producing a final total of nine samples of 150 ng RNA for each sample.

To obtain sufficient RNA for library preparation each RNA pool was cDNA was amplified using the Ovation RNA-Seq system v2 kit (NuGEN) according to the manufacturer's instructions. Library preparation and sequencing was carried out by Edinburgh Genomics (Edinburgh, UK). For each sample, cDNA was converted to a sequencing library using the TruSeq standed mRNA-Seq library for NeoPrep (Ilumina) and barcoded libraries were pooled and sequenced on an Illumina HiSeq 4000 using 125 base paired-end reads to generate 50 million raw reads per sample.

Reads were trimmed using Cutadapt (version 1.9 dev2)[46] for quality at the 3′ end using a quality threshold of 30 and for adapter sequences of the TruSeq Nano DNA kit (AGATCGGAAGAGC) and a minimum length of 35 bp. rRNA reads were removed using sortMeRNA (version 2.1)[47]. The filtered reads were assembled using Trinity (version 2.5)[48], which produced over five million sequences. Transcripts were quantified using the RSEM method[49] and any sequences with TPM (transcripts per million) <1 and isopct (minimal level of dominant isoform expression) <1 were discarded. In order to further reduce any redundancy, transcripts with 95% similarity were clustered using CD-HIT-EST (version 4.7)[50,51]. These contigs were annotated using Trinotate (version 3.1.1)[48]. As part of this pipeline, peptide sequences were predicted using transdecoder, which were further searched against the SwissProt non-redundant database using BLASTP[52]. BLASTX[52] search was also performed with the transcript sequences as the query and the SwissProt non-reduudant database as the target. The Pfam databases[53] were used to predict protein domains using HMMER[54]. SignalP (version 4.1)[55] was used to predict the presence of signal peptides, and TMHMM[56] was used to predict transmembrane helices within the predicted peptide sequences.

The trimmed reads free from rRNA were aligned against the reference transcriptome using bwa mem (version 0.7.13-r1126)[57] with parameter '-M'. Duplicates were marked using Picard tools (version 2.8.1) (http://broadinstitute.github.io/picard). Read counts by transcript were generated using Salmon (version 0.9.1)[58]. The transcriptome assembly produced earlier was used to produce a quasi-mapping index. The quantification step was carried out with parameter '-1 U' to specify an unstranded library and bias correction parameters –seqBias, --gcBias and –posBias[58]. Transcripts were filtered on counts per million (CPM) to remove

transcripts consisting of near-zero counts and to avoid artefacts due to library depth. Transcripts were required to have a CPM >0.3 in at least three samples, corresponding to the smallest sample group as defined by group, once any samples were removed. Reads were normalised using the weighted trimmed mean of $M$-values method[59]. 'TMM' was passed as the method to the calcNormFactors method of edgeR. edgeR (version 3.16.5)[59] was used to perform differential expression analysis. Fold changes were estimated as per the default behaviour of edgeR. Statistical assessment of differential expression was carried out with the quasi-likelihood (QL) $F$-test using the following contrasts: Control versus 1 °C, controls versus 2 °C and 1 °C versus 2 °C. To adjust for confounding covariates such as batch effects, a blocking factor was incorporated as part of the additive model. Differential gene set analysis was carried out using ROAST[60] from the Limma package version 3.30.13 (ref. [61]) of Bioconductor using the same models and contrasts as used in the differential transcript analysis expression. A PCA analysis was undertaken on normalised and filtered expression data to explore observed patterns with respect to experimental factors. The cumulative proportion of variance associated with each factor was used to study the level of structure in the data, while associations between continuous value ranges in principal components and categorical factors was assessed with an ANOVA test. The following Gene Ontology (GO) gene sets were used: GO Molecular Function, GO Cellular Component and GO Biological Process[62]. For GO enrichment analyses, the GO annotations were extracted from the Trans-D_trinotate_annotation_report.txt. ROAST was executed using 9999 rotations. GO terms that were not associated with at least five genes were excluded from the analysis. All transcripts in the contrasts of interest with either a BLASTX and/or a BLASTP annotation were re-searched against the human SwissProt database[63]. A list of unique human protein identifiers was then entered into both the STRING[64] and PANTHER[65] databases to evaluate enrichment of functional groups.

**Upper thermal limit experiments on *Romanchella perrieri*.** UTLs were measured in *R. perrieri* on heated panels from two sites to evaluate whole animal acclimation. The sites were North Cove and South Cove, described earlier. The animals were taken from the same panels in South Cove that were used as a source for *P. stalagmia* for the transcriptome experiments. One heated settlement panel was used for each of the UTL experiments at each temperature, for each site (total of 6 panels), with 25 animals evaluated per panel (total of 150 animals in the experiment). Heated and non-heated panels (one each of control, +1, +2) from the South Cove/aquarium and North Cove sites colonised by *R. perrieri* were transferred to a 60-L jacketed tank with aerated sea water at the same temperature as the ambient sea water (0 °C) and connected to a thermocirculator (Grant Instruments Ltd, Cambridge, UK). The temperature was raised at 1 °C h$^{-1}$ with the temperature limit of each animal noted when they no longer responded to tactile stimuli[35], i.e., did not retract into their exoskeleton, when touched with a dissecting needle seeker. UTL data were non-normal, even after transformations, so non-parametric statistical tests were used to analyse the data. A Mann–Whitney test verified that both the South Cove and North Cove data could be combined ($P = 0.0896$). A Kruskal–Wallis test was used to investigate if there was an effect of temperature on UTL compared with panel treatment and Mann–Whitney tests were subsequently used on these data to identify significance between panel treatments.

**Population genetic analyses.** Transcripts generated for the expression analyses were used in this analysis. Default parameters were used for all programmes unless specified.

Supertranscripts were generated from the assembled transcriptome using the script 'Trinity_gene_splice_modeler.py' provided with Trinity (version 2.5.0)[48]. Trimmed and filtered reads were aligned to the supercontigs using STAR (version 2.5.2b)[66]. Potential PCR duplicates were marked using Picard tools MarkDuplicates. In accordance with GATK[67] best practices, reads with split mappings were split into separate reads in the BAM files using GATK (version 3.7) tool SplitNCigarReads with parameters: -rf ReassignOneMappingQuality -RMQF 255 -RMQT 60 -U ALLOW_N_CIGAR_READS. Local realignment around indels was performed using GATK tools RealignerTargetCreator and IndelRealigner. A single pileup file was generated from the nine BAM files using samtools[68] mpileup (version 1.3) with the parameters '-B -q 20-Q30'. These parameters result in bases with a base quality phred score of less than 30, and an alignment quality of less than 20, being discarded. A single '.sync' file was generated from the pileup file using 'mpileup2sync.jar' from PoPoolation2 (ref. [69]) (version 1.201).

Bayenv2 (ref. [39]) was used to measure the extent to which the allele frequencies of each SNP correlate with temperature. Bayenv2 is designed to take into account the extra level of sampling error arising from pooled data from a small number of individuals. In accordance with recommendations for running Bayenv2, a set of SNPs were selected to generate a matrix of covariance between samples. Only SNPs covered by at least five reads in at least six samples, and with a minor allele supported by at least five reads in total across all samples, were selected, and only one SNP per transcript was included. The covariance matrix was generated using Bayenv2 with the following parameters: -p 9 -k 200000, and specifying specifies four diploid individuals per sample with the '-s' flag. Z-scores for each SNP were calculated using Bayenv2 with the parameters: -p 9 k 200000 -r 8372 -n 1 -e standard_env.txt -x -m pool_matrix.txt –t. Where file 'pool_matrix.txt' contains

the covariance matrix produced in the previous step, and 'standard_env.txt' contains standardised measures of temperature (in degrees centigrade).

Due to the various sources of noise in this dataset, we wanted to filter out SNPs that were most likely affected by sampling error. Before attempting to calculate FDRs or perform GSEA, the SNPs were filtered to remove low coverage SNPs. Specifically, we removed SNPs that were not covered by at least five reads in each sample, or which had a minor allele supported by less than 15 reads in total across the nine samples. Because Bayenv2 does not produce $P$ values, we estimated the statistical significance of our results by reference to a null distribution of Z-scores. This was created by randomly permuting the labels in file 'standard_env.txt' 100 times and recalculating the Z-scores for each SNP. The null distribution of Z-scores allowed us to calculate the probability of a high Z-score arising by chance. The false discovery rate (FDR) for each SNP was then calculated as follows: FDR = $ip/n$, where $i$ = number of SNPs in the dataset that achieved an equal or greater Z-score, $n$ = total number of SNPs in all null permutations that achieved an equal or greater than Z-score, and $p$ = number of permutations.

A score for each supertranscript was calculated by taking the mean Z-score of all SNPs from that supertranscript. FDRs for each supertranscript with more than five SNPs that passed the filter were calculated from the null distribution as follows: FDR = $ip/n$, where $i$ = number of supertranscripts in dataset that achieved an equal or greater mean Z-score, $n$ = total number of supertranscripts with at least five SNPs passing the filter in all null permutations that achieved an equal or greater mean Z-score, and $p$ = number of permutations. For each SNP, the minor allele frequency (MAF) was calculated for each of the three groups of samples. The MAF was calculated for each individual sample as follows: MAF = $m/t$, where $m$ = number of reads supporting minor allele and $t$ = total number of reads covering site. The effective allele number was also calculated for each sample at each SNP, using the formula $e = ((nc)-1)/(n+c)$. MAF was calculated for each of the three groups by taking the average MAF for each sample weighted by the effective allele number for that sample. This allows varying the sampling error arising from the varying depths of coverage between samples to be accounted for.

**Biofilm methods.** Panels from North Cove were brought up to the surface by SCUBA divers and placed in a 10 L tank on the boat with sea water at the same temperature as the ambient sea water (~0 °C). Biofilm swabs were taken from the panels while on the boat and stored in 100% ethanol for subsequent analyses. Four to five biofilm swabs were taken per treatment (control, +1, +2). Total genomic DNA was extracted from the biofilm swabs using the PowerBiofilm DNA isolation kit (MO BIO Laboratories, Inc.) following the manufacturer's instructions. A blank swab was also included and extracted as a no-template control. DNA samples were assessed for concentration and quality using a NanoDrop ND-100 Spectrometer (NanoDrop Technologies) and an Agilent 2200 Tapestation (Agilent Technologies). To enable compatibility with the Illumina 16S Metagenomic Sequencing Protocol, the hypervariable V4 region of the 16S rRNA genes was PCR-amplified using the 16S Amplicon PCR forward primer (5′-TCGTCGGCAGCGTCAGATG TGTATAAGAGACAGCCTACGGGNGGCWGCAG) and the 16S Amplicon reverse PCR primer (5′-GTCTCGTGGGCTCGGAGATGTGTATAAGAGACAGG ACTACHVGGGTATCTAATCC). PCR reactions were carried out using the KAPA Hifi HotStart ReadyMix according to manufacturer's instructions. Amplifications were carried out in an AlphaCycler (PCRmax) under the following conditions: 95 °C for 30 s, followed by 25 cycles of 95 °C for 30 s, 55 °C for 30 s and 72 °C for 30 s. A final elongation step at 72 °C for 5 min was performed. Resulting PCR products were checked by standard agarose gel (1.5%) electrophoresis and purified using the QIAquick PCR Purification kit (Qiagen) following the manufacturer's instructions.

Library preparation and sequencing was carried out by the Department of Biochemistry at the University of Cambridge. For each sample ($n = 4$–5 swabs per sample per treatment: control, +1, +2 and control) DNA was converted in to a sequencing library using the 16S Metagenomic Sequencing Library preparation kit (DNA input 1 ug, 8 PCR cycles), and sequenced in triplicate (39 samples total) on an Illumina MiSeq using 300 base paired-end reads, to generate 44–50 million raw reads per pool.

Oligotyping analysis was used[22] to define biofilm community composition differences in heated and non-heated settlement panels. Adapters were trimmed from the raw reads using trim_galore software (https://www.bioinformatics. babraham.ac.uk/projects/trim_galore/). Reads were merged using mothur v.1.35.1 (ref. [70]), MiSeq SOP site accessed on the 18/08/17. Entropy and oligotyping analyses were conducted according to ref. [21]. Sequences were not aligned to a reference alignment since length read varied due to partially overlapping reads. Hence the *0-pad-with-gaps* script from the Minimum entropy decomposition (MED) pipeline 1.2 (ref. [71]) was run on the reads. All analyses were carried out using default parameters unless otherwise specified. MED analysis was performed using the MED pipeline version 1.2. The minimum substantive abundance criterion (M) was set to 50 to filter noise in the data. After the initial round of oligotyping, high entropy positions were chosen (-C option). To minimise the impact of sequences errors, an oligotype was required to be represented in at least 1000 reads (-M option). Moreover, rare oligotypes present in less than five samples were discarded (-s option). These parameters led to 1,478,129 sequences left in the database. Oligotypes were searched using blastn against the ENA sequence database using NCBI Blast +via the EBI web services[72]. Further oligotyping

analysis was performed on the rare oligotypes removed from the original analysis to look at differences in microbial diversity in the rarer species between treatments (control, +1, +2). Rare oligotypes were explored by selecting the parameter M (minimum substantive abundance) and by using the oligotype command in the pipeline to scrutinise the data. The minimum substantive abundance criterion was set to 50 (-M 50) to filter noise in the data. After the initial round of oligotyping, nucleotide positions (9, 13, 120, 122 and 130) were carefully selected to explore the rare oligotypes. Two more rounds of oligotyping were performed with the minimum substantive abundance set to 50 and by further selecting nucleotide positions (9, 13, 26, 80, 120, 122 and 130). Those rare oligotypes enriched by PCR errors did not converge. Individual oligotype sequences were searched against reference sequences in the NCBI's nr database using BLAST[73].

**Reporting Summary**. Further information on research design is available in the Nature Research Reporting Summary linked to this article.

## Data availability

The EMBRIC configurator data management service [https://doi.org/10.7490/f1000research.1116538.1] was used for advice on data coordination and standards. The transcriptome data are available from the European Nucleotide Archive with the ENA accession number: PRJEB27537. The biofilm data are also available from the European Nucleotide Archive with the ENA accession number: PRJEB30562. The source data underlying Figs. 1, 2, and 3a are provided as a Source Data file. The data underlying Fig. 2 are also available from the UK Polar Data Centre, Natural Environment Research Council, UK Research & Innovation: https://doi.org/10.5285/93eaaf9e-0624-441b-81f0-0438b844f6bb.

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

## Acknowledgements

The work was funded by NERC standard grant NE/J007501/1 awarded to L.S.P. which included a tied studentship awarded to L.V.N. and access to the Edinburgh facility (project references: 10742 and 11367). This work was also supported by a Deutsche Forschungsgemeinschaft (DFG) grant to J.I.H. under project number HO 5122/7-1. The transcriptome Illumina sequencing and analyses were carried out by Edinburgh Genomics, The University of Edinburgh. Edinburgh Genomics is partly supported through core grants from NERC (R8/H10/56), MRC (MR/K001744/1) and BBSRC (BB/J004243/1). The biofilm amplicon sequencing was performed at the University of Cambridge, Department of Biochemistry Sequencing Facility. The authors would like to thank Mark Preston for his work on the development of the heated panels, the Rothera marine teams from 2013 to 2016 for help with maintenance and photography of the panels, especially Simon Morley and David Barnes and Ben Temperton at the University of Exeter for his advice on the biofilm analyses.

## Author contributions

M.S.C. helped devise the project, supervised the transcriptome analyses, plus performed the biological annotation and supervised the student. L.V.N. collected the samples, extracted and amplified the RNA for the transcriptomes analyse and performed the biofilm oligotyping and amplicon database submission. J.I.H. supervised the population genetic analyses. A.J.D. advised on the analyses and supervised the student. U.H.T. performed the transcriptome analyses and transcriptome database submission. F.T. performed the population genetic analyses. G.V.A. ran the panels and advised on analyses. L.S.P. devised the original project, obtained the funding, helped to run the panels, advised on the analyses and supervised the student. M.S.C. led the writing of the paper and all other authors contributed to writing the manuscript and have approved the final version.

## Additional information

**Competing interests:** The authors declare no competing interests.

