## [Peer Review File · Nature Communications]

Reviewers' Comments:

Reviewer #1:

Remarks to the Author:

This paper describes a study based around a long term deployment of a heated plate technology for assessing the response of encrusting communities to warming in the Antarctic. It is worth noting that I have no experience of oligotype analysis (which forms a small part of the analysis), but I have reviewed the paper and have no concerns about scientific rigour. I think that this paper is an important contribution and certainly of topical relevance to Nature Communications and the broader scientific community. I have made various suggestions for small edits and encourage the authors to act on these wherever possible.

Specific comments to authors:

Line 87 – A typo in this sentence – is the upper limit of this range 0 or 1? "Panels comprised controls (non-heated, experiencing ambient temperatures roughly between -2°C and +1 0°C)"

Line 98 – Provide the spirobid species name in the Introduction – now I have read the paper I see why this was not done so please ignore this comment.

Line 119 – Provide full species name on first mention

Line 120 – Be clear that the assembly was performed on the raw reads from all samples (assuming this was the case?).

Line 119-120 – The authors write 'Raw assembly of the reads produced over 5 million contigs greater than 100 bp that contained over 4.28 Gb of data (Supplementary Table S1)'. Do they mean assembly, or are these just the raw reads? Also, they write 'A series of filtering steps....' – please provide this detail, if not in the manuscript, then as Supplementary.

Line 129 – Not clear how the DE analysis links to the PCA. The PCA sentence seems out of place and I suggest it would be better to move the detail of the numbers of genes up and down regulated up to follow this first sentence. Perhaps having the PCA results either preceding or following on from the DE analysis would be clearer?

What was the PCA on – are these counts, or the DE scores?

Figure 1 legend – be clear about what the PCA was performed on. Can anything informative be added about the labels in the legend? Small point – but 'degree' and 'degrees' are both used and really need replacing with °C. R code: `expression(0~degree~C)`

Line 140-143 – Interesting: I would lead with the second sentence to make clear the direction of the enrichment and up-regulation at the higher temperature as currently this could be interpreted backwards, with the second sentence tacked on.

Line 150 – A missing end to the sentence.

Line 166 – Why are these data not shown? If you discuss them in the paper surely you need to present them in some form at least, even if just as supplementary.

Line 173 – Please tell us the species used in place of *P. stalagmia* in this introductory sentence. Also, a brief explanation of how this was done – were they also on the same experimental panels and for the

same duration? Or taken from different panels?

Line 178-179 – Can you provide some indication of variation or range in these measures of UTL?

Line 188 – Unclear how this collective analysis was done? Can they provide more detail or a supporting reference.

In the Table legends be clear that these are for the spirorbids

The fonts in Figure 3 are very small and difficult to read.

Reviewer #2:

Remarks to the Author:

In this manuscript, Clark et al. test the ability of Antarctic encrusting organisms to perform under small (+1 and +2C) increases in substrate temperature. Like the earlier paper by the same group (Ashton et al. 2017), which used the same panels and organisms to assess the thermal sensitivity of growth, I think this is an exciting project and this MS adds population genetic analyses and transcriptomic and behavioral tests for physiological stress to the growth information. There are several areas where the MS could be improved, however, particularly in methods and the physiological and evolutionary interpretations.

Perhaps my most relevant comments have to do with the physiological (UTL) experiments and their interpretations. First, the methods are not described in enough detail; there are no sample sizes for the number of animals tested and the number of panels tested is not very clear, nor is it clear how these are divided between the two sites (North Cove and South Cove/Aquarium), or how it was determined whether or not animals “responded to tactile stimuli” (the reference given is general). Second, I can’t quite put together the conclusion that spirorbids on the +2 panels were in a “permanent state of resistance and/or decline” (which seems like a reasonable interpretation of the transcriptome and UTL data) with the very rapid growth rates of the same taxa on the same panels in the previously published paper (). Is the implication that the animals can grow rapidly under the warming conditions for a long time, but can’t sustain it indefinitely? Is there a seasonal component, such that there was abundant food during the early months allowing for rapid growth, but metabolism could not be sustained at elevated temperatures in the low-food months? Or could this be an effect of the natural life span of the animals, such that individuals on the warmed panels were actually physiologically older than individuals on the cool panels? That raises the possibility that the transcriptome data may be capturing a signal of natural senescence. More clarity and discussion of these points would strengthen the MS.

Another general comment is that the MS claims to find little evidence for “genetic adaptation” within the spirorbid community (line 344). I agree that the lack of genetic differentiation between temperature treatments provides no evidence for post-recruitment selection, but that is distinct from adaptation as is generally defined. In this setup, unless the spirorbids have very low dispersal and self-recruit onto the same panel as the parents, all recruits must come from the outside and so even if there were differentiation between plates, while it would suggest differential survival of genotypes arriving on the plates, it would not show adaptation to new conditions.

Other comments:

(1) The description of panel outplant and collection is not entirely clear. Numbers of panels at the two

sites are not given; similarly, while "one set" of panels (North Cove) was in the field for the entire 18 months, the other "set" (three?) was held in a "mesocosm" at the "same environmental conditions as the original site" for the last 9 months. This needs more explanation, because mesocosms do not replicate real environmental conditions very well, particularly if the mesocosm was an aquarium system where even if the temperature treatments were maintained perfectly, many things would be different like water chemistry, food availability, recruitment, competition, etc. The introductory claim that 18 months is the "longest in situ experimental manipulation of temperature anywhere in the oceans to date" is a little bit misleading since it appears (though is not entirely clear) that while some of the data come from 18-month field outplants, something like half of it comes from organisms that were in the laboratory for the last 9 of those 18 months.

(2) In the abstract and elsewhere, the MS emphasizes that Antarctic fauna have poor capacity for adaptation in the face of climate change because of their long generation times and extended lifespans. The study taxa in the MS, however, are microbes and spirorbid polychaetes, the latter of which are small-bodied and fast-growing compared to other metazoans. Thus, while experimentally practicable in a way most Antarctic metazoans are not, they probably represent a comparatively high-energy lifestyle and may not be the best group to shed light on the future of larger taxa. Also, as a minor point, the phrase "post-recruitment selection" is not quite accurate; in this setup there would be no way to establish whether selection, if it happened, occurred before or after settlement.

(3) I suggest a couple of changes to terminology: "catholic diet" to something more informative (does this just mean non-filter-feeders?); "Clearly, the $_1$ animals were still trying to acclimatize their physiologies to the warmer conditions, even after 18 months" (lines 289-290) to a more objective statement that does not conflict with the "permanent state of resistance and/or decline" that the animals were in.

Reviewer #3:

Remarks to the Author:

This is an important topic and one which I've studied in the past. However, most of the results relate to molecular methods with which I'm not particularly familiar. My comments are therefore really those of an interested marine biologist rather than a real subject specialist.

I realise that there was a page limit, but the amount of basic information to set the study up was limiting in my opinion. It wasn't clear to me whether the panels were put in clean at the start of the 18 month study, or whether these panels had already been colonised and were then warmed up? This makes a big difference to the implications because in one scenario we're talking about pure acclimation, whereas in the other we've got the effects of developing under different conditions. As the paper is supposed to be about acclimation I'd have thought this meant that the communities already existed and then were randomly allocated to the different temperature treatments, but as no details are given I'm assuming that in fact this was an experiment where the assemblages formed under different conditions.

The introduction is very general on the topic of future climate, WAP etc. What I really needed in order to understand this experiment was information about the previous findings on community/assemblage formation under different temperature treatments. What mysteries did this throw up which needed explanation using the present experiment? The context just isn't developed well enough for me. This is explained somewhat in the Discussion, but this is too late in my opinion, and means that the reasons for doing the work aren't really clear when reading the paper.

There is an absence of practical details. When the methods are first described they don't even say how many panels there are in each treatment (from Fig 1 I guess 3?). Then later there's a reference to North Cove and South Cove panels, but were these the only locations? When "sets" of panels are referred to what does this mean? One panel of each temperature? How were these all set up? Alternating control, +1, +2, control....etc?

For the description of the UTL experiments the panels used were from "...North Cove and a set of panels from South Cove". So is this all the North Cove panels and some of the South cove ones? This is just all unnecessarily confusing.

This isn't a conventional method for testing upper thermal limits, it's more of a critical behaviour test, could you please clarify this?

What was the size of the different worms? They'd spent their whole lives at different temperatures (I assume?) or at least the last 18 months so had the warmer ones grown faster or slower? Do you know anything about their growth rate in summer and winter? Their temperature tolerances would be affected by their body mass, so maybe controlling for this would give different answers? Even if total size now is similar, if animals under warmer conditions has more temporally variable growth this could be relevant to their stress responses - effectively like catch-up growth does. Was this considered? Doesn't necessarily invalidate your findings, but does need to be thought about I think.

Page 5, please move the results into the Results section.

Also on page 5, an absence of difference in UTL is compared to differences in bacterial community composition - pretty different metrics. In fact, given the topic of the paper, I don't really know why the bacterial data are even in this paper. They'd be a better comparison in a paper on invertebrate community composition. Is acclimation measured in the bacteria or is community composition not changing being used to infer acclimation? The mechanism might be more evolutionary, depending on how many generations had passed, it's just not the same process.

The paper title implies analysis of a range of encrusting animals, when in fact it's UTL in one species and molecular measurement of stress in another. It takes a long time for this to become clear though. Figure 2 gives the species in question, Figure 1 does not, but searching through the results shows that fig 1 is a different species from fig 2.

My feeling is that there's enough material for two small papers here, but the microbial and worm data don't really fit together well and makes the story more confusing than it needs to be.

The absence of important information, and/or placing of important information well after it was needed to explain the motivation and value of the paper, need to be fixed.

I can definitely see the value of the work, but in my opinion it needs major editing before publication.

Reviewers' comments:

Reviewer #1 (Remarks to the Author):

This paper describes a study based around a long term deployment of a heated plate technology for assessing the response of encrusting communities to warming in the Antarctic. It is worth noting that I have no experience of oligotype analysis (which forms a small part of the analysis), but I have reviewed the paper and have no concerns about scientific rigour. I think that this paper is an important contribution and certainly of topical relevance to Nature Communications and the broader scientific community. I have made various suggestions for small edits and encourage the authors to act on these wherever possible.

Specific comments to authors:

Line 87 – A typo in this sentence – is the upper limit of this range 0 or 1? “Panels comprised controls (non-heated, experiencing ambient temperatures roughly between -2°C and +1 0°C)”

Apologies, this was +1°C and has now been corrected

Line 98 – Provide the spirobid species name in the Introduction – now I have read the paper I see why this was not done so please ignore this comment.

Many thanks: in line with your suggestion, we have ignored this comment.

Line 119 – Provide full species name on first mention

corrected

Line 120 – Be clear that the assembly was performed on the raw reads from all samples (assuming this was the case?).

This is indeed the case: see below for revised sentence

Line 119-120 – The authors write ‘Raw assembly of the reads produced over 5 million contigs greater than 100 bp that contained over 4.28 Gb of data (Supplementary Table S1)’. Do they mean assembly, or are these just the raw reads? Also, they write ‘A series of filtering steps....’ – please provide this detail, if not in the manuscript, then as Supplementary.

These details are described fully in the Materials and methods section. This sentence has now been amended as “Assembly of the raw reads from all samples produced over 5 million contigs greater than 100 bp that contained over 4.28 Gb of data (Supplementary Table S1). A series of filtering steps (described in Materials and methods: Transcriptome methods: Bioinformatic analyses) provided a reference transcriptome containing 61,421 high quality "transcripts" with the mean length of 486bp and a total size of 29.92 Mb.”, which incorporates the previous query.

Line 129 – Not clear how the DE analysis links to the PCA. The PCA sentence seems out of place and I suggest it would be better to move the detail of the numbers of genes up and down regulated up to follow this first sentence. Perhaps having the PCA results either preceding or following on from the DE analysis would be clearer?

OK, good point! we have moved the sentences around and now lead with the differential expression data.

What was the PCA on – are these counts, or the DE scores?

This was on DE scores and a note added to this effect. The following has been added to the Materials and methods section “A PCA analysis was undertaken on normalised and filtered expression data to

explore observed patterns with respect to experimental factors. The cumulative proportion of variance associated with each factor was used to study the level of structure in the data, while associations between continuous value ranges in principal components and categorical factors was assessed with an ANOVA test.”

Figure 1 legend – be clear about what the PCA was performed on. Can anything informative be added about the labels in the legend? Small point – but ‘degree’ and ‘degrees’ are both used and really need replacing with °C. R code: `expression(0~degree~C)`

This legend has now been changed to “Figure 1: PCA plot on normalised and filtered expression data of the different treatments (control, +1, +2)” and the key on the figure has been changed. We have also added further information in the main text.

Line 140-143 – Interesting: I would lead with the second sentence to make clear the direction of the enrichment and up-regulation at the higher temperature as currently this could be interpreted backwards, with the second sentence tacked on.

Good point, these have been changed round.

Line 150 – A missing end to the sentence.

Apologies, this is a wording issue, this sentence has been modified to: “Given the lack of any GO enrichment for particular categories of gene functions and the large number of transcripts involved, more detailed analyses concentrated on the expression profiles of the +2 samples examining the annotation of the up-regulated transcripts.”

Line 166 – Why are these data not shown? If you discuss them in the paper surely you need to present them in some form at least, even if just as supplementary.

Apologies, these have now been added as Supplemental Figure S1.

Line 173 – Please tell us the species used in place of *P. stalagmia* in this introductory sentence. Also, a brief explanation of how this was done – were they also on the same experimental panels and for the same duration? Or taken from different panels?

Apologies, the species was actually in the sub-title of the section, but we have now added the name in full in the sub-title and also in brackets in the appropriate place. These were taken from the same panels and subject to the same experimental regime as the *P. stalagmia*: these points have now been added to the methods with a note in this section as well.

Line 178-179 – Can you provide some indication of variation or range in these measures of UTL?

Of course: this has now been altered to mean temperature \pm SE mean in the text, as Figure 1 shows the median and range in % quartiles.

Line 188 – Unclear how this collective analysis was done? Can they provide more detail or a supporting reference.

More description has been added: The relevant sentence has now been altered to “However, when the mean Z-scores of SNPs across genes were assessed, 91 out of 521 genes (17.5%) showed a significant association with temperature after FDR correction (Table 2).”

In the Table legends be clear that these are for the spioribids

This has now been made clear with the species tacked onto the end of the table legends.

The fonts in Figure 3 are very small and difficult to read.

Agreed, they do look small. However, we think some of the problem is due to the reproduction in word. These files will be supplied as high quality .tiff files and we can discuss this with the production team and whether we need to make changes.

Reviewer #2 (Remarks to the Author):

In this manuscript, Clark et al. test the ability of Antarctic encrusting organisms to perform under small (+1 and +2C) increases in substrate temperature. Like the earlier paper by the same group (Ashton et al. 2017), which used the same panels and organisms to assess the thermal sensitivity of growth, I think this is an exciting project and this MS adds population genetic analyses and transcriptomic and behavioral tests for physiological stress to the growth information. There are several areas where the MS could be improved, however, particularly in methods and the physiological and evolutionary interpretations.

Perhaps my most relevant comments have to do with the physiological (UTL) experiments and their interpretations. First, the methods are not described in enough detail; there are no sample sizes for the number of animals tested and the number of panels tested is not very clear, nor is it clear how these are divided between the two sites (North Cove and South Cove/Aquarium)

A more extensive explanation has now been added to the materials and methods section, including panel design and numbers of animals.

, or how it was determined whether or not animals “responded to tactile stimuli” (the reference given is general).

Again, more detail has been added to the materials and methods

Second, I can't quite put together the conclusion that spirorbids on the +2 panels were in a “permanent state of resistance and/or decline” (which seems like a reasonable interpretation of the transcriptome and UTL data) with the very rapid growth rates of the same taxa on the same panels in the previously published paper (). Is the implication that the animals can grow rapidly under the warming conditions for a long time, but can't sustain it indefinitely? Is there a seasonal component, such that there was abundant food during the early months allowing for rapid growth, but metabolism could not be sustained at elevated temperatures in the low-food months?

This is exactly our point, but we obviously did not state it clearly enough in the text and so have tried to clarify it throughout the revised text.

Or could this be an effect of the natural life span of the animals, such that individuals on the warmed panels were actually physiologically older than individuals on the cool panels? That raises the possibility that the transcriptome data may be capturing a signal of natural senescence. More clarity and discussion of these points would strengthen the MS.

This is highly unlikely for several reasons: these animals can live 4-5 years (Barnes, pers comm) and in a set of control panels retrieved after 2.5 years, all the spirorbids were still alive. Also the expression profiles are not characteristic of natural senescence, which would involve a general down-regulation of metabolism and gene expression, which was not seen in this study. In addition the types of genes differentially expressed (involving the stress response etc) are not symptomatic of natural senescence. In line with the request of this reviewer, we have added a note to this effect in the text and hope that overall our message is much clearer.

Another general comment is that the MS claims to find little evidence for “genetic adaptation” within the spirorbid community (line 344). I agree that the lack of genetic differentiation between

temperature treatments provides no evidence for post-recruitment selection, but that is distinct from adaptation as is generally defined. In this setup, unless the spirorbids have very low dispersal and self-recruit onto the same panel as the parents, all recruits must come from the outside and so even if there were differentiation between plates, while it would suggest differential survival of genotypes arriving on the plates, it would not show adaptation to new conditions.

Many thanks: absolutely correct. In the two instances in the text where we have mentioned adaptation, this has been changed to differential survival of genotypes.

Other comments:

(1) The description of panel outplant and collection is not entirely clear. Numbers of panels at the two sites are not given; similarly, while “one set” of panels (North Cove) was in the field for the entire 18 months, the other “set” (three?) was held in a “mesocosm” at the “same environmental conditions as the original site” for the last 9 months. This needs more explanation, because mesocosms do not replicate real environmental conditions very well, particularly if the mesocosm was an aquarium system where even if the temperature treatments were maintained perfectly, many things would be different like water chemistry, food availability, recruitment, competition, etc.

Agreed, this is unclear and the description was originally kept very brief for space reasons. We have now expanded on the experimental design in the materials and methods section. We have also added in more information on the aquarium system: which a flow-through system and has an unfiltered water intake at 7M. Therefore we are confident that the water chemistry and food supply was the same for the animals in the aquarium compared with those in the sea (especially considering that most of these 9 months comprised the winter period, which has very low levels of phytoplankton in the water). It should be noted that in this experiment we were not studying either recruitment or competition.

The introductory claim that 18 months is the “longest *in situ* experimental manipulation of temperature anywhere in the oceans to date” is a little bit misleading since it appears (though is not entirely clear) that while some of the data come from 18-month field outplants, something like half of it comes from organisms that were in the laboratory for the last 9 of those 18 months.

Agreed. In the Abstract, this has now been changed to “we show that after 18 months (involving some of the longest *in situ* experimental manipulations of temperature in the ocean to date)” and we have clarified the experimental detail further in the main text.

(2) In the abstract and elsewhere, the MS emphasizes that Antarctic fauna have poor capacity for adaptation in the face of climate change because of their long generation times and extended lifespans. The study taxa in the MS, however, are microbes and spirorbid polychaetes, the latter of which are small-bodied and fast-growing compared to other metazoans. Thus, while experimentally practicable in a way most Antarctic metazoans are not, they probably represent a comparatively high-energy lifestyle and may not be the best group to shed light on the future of larger taxa.

These animals live 4-5 years (Barnes, pers comm), which is much longer than similar species elsewhere in the world and in fact represent, for these taxa, a low energy life-style. They are used to demonstrate the potential response of filter feeders and the issue of seasonality in that response, which will also affect much larger animals. The responses of the spirorbids are also used as proxies for the other species on the panel, such as ascidians and bryozoans, which also showed massive increases in growth rates on these panels and again, live much longer in the Antarctic than elsewhere. We have amended our main text to incorporate these comments from the reviewer.

Also, as a minor point, the phrase “post-recruitment selection” is not quite accurate; in this setup there would be no way to establish whether selection, if it happened, occurred before or after settlement.

Agreed this has been removed or changed to differential survival between genotypes.

(3) I suggest a couple of changes to terminology: “catholic diet” to something more informative (does this just mean non-filter-feeders?); “Clearly, the _1 animals were still trying to acclimatize their physiologies to the warmer conditions, even after 18 months” (lines 289-290) to a more objective statement that does not conflict with the “permanent state of resistance and/or decline” that the animals were in.

Exactly, this has now been changed from catholic diets to non-filter-feeders. The sentence on acclimatisation has been changed to “. Thus, the +1 animals were still unable to acclimate their physiologies to the warmer conditions, even after 18 months.” Which fits much better with the permanent state of resistance statement.

Reviewer #3 (Remarks to the Author):

This is an important topic and one which I've studied in the past. However, most of the results relate to molecular methods with which I'm not particularly familiar. My comments are therefore really those of an interested marine biologist rather than a real subject specialist.

I realise that there was a page limit, but the amount of basic information to set the study up was limiting in my opinion. It wasn't clear to me whether the panels were put in clean at the start of the 18 month study, or whether these panels had already been colonised and were then warmed up? This makes a big difference to the implications because in one scenario we're talking about pure acclimation, whereas in the other we've got the effects of developing under different conditions. As the paper is supposed to be about acclimation I'd have thought this meant that the communities already existed and then were randomly allocated to the different temperature treatments, but as no details are given I'm assuming that in fact this was an experiment where the assemblages formed under different conditions.

Apologies that this was not made clear. The following sentence has now been added to the methods “At the start of the 18 month deployments, all panels were brand new, placed on site and then gradually warmed up to the relevant temperature for colonisation *in situ*.”. The methods with regard to panel design has also been expanded.

The introduction is very general on the topic of future climate, WAP etc. What I really needed in order to understand this experiment was information about the previous findings on community/assemblage formation under different temperature treatments. What mysteries did this throw up which needed explanation using the present experiment? The context just isn't developed well enough for me. This is explained somewhat in the Discussion, but this is too late in my opinion, and means that the reasons for doing the work aren't really clear when reading the paper.

Thank you for this comment. When we were writing the paper, whether to put the original study more prominently in the introduction was a big debate amongst the authors: some felt it should be positioned as a study in its own right, with the original experiment only described in the discussion to avoid detracting from the study presented here, whilst others wanted something more up-front. In line with this reviewers comments, we have now added a very short introduction to the original study findings in the introduction, as well as leaving the more extensive description in the discussion.

There is an absence of practical details. When the methods are first described they don't even say how many panels there are in each treatment (from Fig 1 I guess 3?). Then later there's a reference to North Cove and South Cove panels, but were these the only locations? When "sets" of panels are referred to what does this mean? One panel of each temperature? How were these all set up? Alternating control, +1, +2, control....etc?

Agreed, more detail should have been added, but was originally left out for brevity. This section in the materials and methods has now been greatly expanded.

For the description of the UTL experiments the panels used were from "...North Cove and a set of panels from South Cove". So is this all the North Cove panels and some of the South cove ones? This is just all unnecessarily confusing.

Agreed, this does seem a bit confusing. This part of the materials and methods has been changed, with much more information added about the experimental design with regard to the heated settlement panels.

This isn't a conventional method for testing upper thermal limits, it's more of a critical behaviour test, could you please clarify this?

We disagree with this reviewer, as ramping experiments to determine UTL as a measure of whole animal acclimation have been used in numerous studies such as:

- Fry, F. E. J., Brett, J. R. and Clawson, G. H. (1942). Lethal limits of temperature for young speckled trout (*Salvelinus fontinalis*). University of Toronto Studies, Biological Series, number 54. *Publ. Ontario Fish. Res. Lab.* **72**, 1-79.
- Prosser, C. L. (1973). *Comparative Animal Physiology*, 3rd edn, pp. 966. Philadelphia, PA: Saunders.
- Schmidt-Nielsen, K. (1997). *Animal Physiology: Adaptation and Environment*, 4th edn, Cambridge: Cambridge University Press, 602 pp.
- Pörtner, H. O., Peck, L. S. and Somero, G. A. (2007). Thermal limits and adaptation in marine Antarctic ectotherms: an integrative view. *Philosophical Transactions of the Royal Society. B* **362**, 2233-2258.
- Peck, L. S., Clark, M. S., Morley, S. A., Massey, A. & Rossetti, H. (2009). Animal temperature limits and ecological relevance: effects of size, activity and rates of change. *Functional Ecology* **23**, 248-256.
- Terblanche JS, Hoffmann AA, Mitchell KA, Rako L, Le Roux PC, Chown SL (2011). Ecologically relevant measures of tolerance to potentially lethal temperatures. *Journal of Experimental Biology*, **214**, 3713–3725.
- Bilyk, K. T. and DeVries, A. L. (2011). Heat tolerance and its plasticity in Antarctic fishes. *Comparative Biochemistry and Physiology*. **158A**, 382-390.
- Peck, L. S., Morley, S. A., Richard, J. & Clark, M. S. (2014) Acclimation and thermal tolerance in Antarctic marine ectotherms. *Journal of Experimental Biology* **217**, 16-22.

What was the size of the different worms? They'd spent their whole lives at different temperatures (I assume?) or at least the last 18 months so had the warmer ones grown faster or slower? Do you know anything about their growth rate in summer and winter? Their temperature tolerances would be affected by their body mass, so maybe controlling for this would give different answers? Even if total size now is similar, if animals under warmer conditions has more temporally variable growth this could be relevant to their stress responses - effectively like catch-up growth does. Was this considered? Doesn't necessarily invalidate your findings, but does need to be thought about I think. There are data on growth rates in the different seasons in the previous paper (Ashton et al., (2017) *Current Biology*). In our previous work we found no difference in UTL across different sized mature adults in a range of Antarctic species (Peck et al., (2009) *Functional Ecology*). The reviewer is correct that the size of the animals used in the different treatments could be different because of different growth rates, but all animals tested were mature adults and therefore any differences in response due to size would be minimal and very difficult to detect.

Page 5, please move the results into the Results section.

We totally understand this comment from the reviewer, however on the advice of one of the co-authors, who has published recently in Nature Comms and reading many recent articles, it appears to be Nature Comms style to add a small section of “we show here that....” “Here we demonstrate that....” and paraphrase the results in the introduction, before the results section. We are therefore conforming to journal style.

Also on page 5, an absence of difference in UTL is compared to differences in bacterial community composition - pretty different metrics. In fact, given the topic of the paper, I don't really know why the bacterial data are even in this paper. They'd be a better comparison in a paper on invertebrate community composition. Is acclimation measured in the bacteria or is community composition not changing being used to infer acclimation? The mechanism might be more evolutionary, depending on how many generations had passed, it's just not the same process.

The reviewer is correct that we are using community composition as a proxy for acclimation, which given their very short life spans is a different process: most likely evolutionary. We have made this clearer in the main text. We also refer the reviewer to the comment of the handling editor and also our responses to their last question as to why we have included the bacterial data here. We think the contrast between prokaryotes and eukaryotes makes an important point and adds to the overall study. The other two reviewers have not suggested separating the two out.

The paper title implies analysis of a range of encrusting animals, when in fact it's UTL in one species and molecular measurement of stress in another. It takes a long time for this to become clear though.

We are using the spirorbids as “model” species and proxys for other species on the panels, as all the major colonisers showed the same response of rapid growth in the Ashton et al., (2017) paper. We have now made this clearer in the introductory text, including the fact that spirorbids were used because they provided sufficient material for RNA studies and were present in sufficient numbers to enable replication in those studies.

Figure 2 gives the species in question, Figure 1 does not, but searching through the results shows that fig 1 is a different species from fig 2.

This has now been amended in line with a previous reviewer's comments: species names are in the figure titles.

My feeling is that there's enough material for two small papers here, but the microbial and worm data don't really fit together well and makes the story more confusing than it needs to be.

On the advice of the handling editor, we have kept this as a single paper, plus see comments above. This study results from a NERC grant with a tied studentship (with this study, the work of the student). In the applications that led to funding, the main query with regard to the student project was how does the biofilm differ compared to the encrusting fauna: grant referees wanted to see both elements together and we think this adds further support for including the microbial analyses alongside the acclimation study.

Reviewers' Comments:

Reviewer #1:

Remarks to the Author:

I am happy with the changes that the authors have made in response to the previous round of review and feel that all points of concern have been adequately addressed.

Reviewer #2:

Remarks to the Author:

The authors did a good job addressing the concerns I had about the first version, and I have no additional comments.